

SciPost Phys. 1(1), 008 (2016)

# Correlations of zero-entropy critical states in the XXZ model: integrability and Luttinger theory far from the ground state

**R. Vlijm, I.S. Eliëns and J.-S. Caux**

Institute for Theoretical Physics, University of Amsterdam,
Science Park 904, 1098 XH Amsterdam, The Netherlands

## Abstract

Pumping a finite energy density into a quantum system typically leads to 'melted' states characterized by exponentially-decaying correlations, as is the case for finite-temperature equilibrium situations. An important exception to this rule are states which, while being at high energy, maintain a low entropy. Such states can interestingly still display features of quantum criticality, especially in one dimension. Here, we consider high-energy states in anisotropic Heisenberg quantum spin chains obtained by splitting the ground state's magnon Fermi sea into separate pieces. Using methods based on integrability, we provide a detailed study of static and dynamical spin-spin correlations. These carry distinctive signatures of the Fermi sea splittings, which would be observable in eventual experimental realizations. Going further, we employ a multi-component Tomonaga-Luttinger model in order to predict the asymptotics of static correlations. For this effective field theory, we fix all universal exponents from energetics, and all non-universal correlation prefactors using finite-size scaling of matrix elements. The correlations obtained directly from integrability and those emerging from the Luttinger field theory description are shown to be in extremely good correspondence, as expected, for the large distance asymptotics, but surprisingly also for the short distance behavior. Finally, we discuss the description of dynamical correlations from a mobile impurity model, and clarify the relation of the effective field theory parameters to the Bethe Ansatz solution.


## 1  Introduction

The ground states and low-lying excitations of one-dimensional many-body quantum systems often display interesting features associated to the Tomonaga-Luttinger liquid universality class [1, 2], notable examples being one-dimensional quantum gases and spin chains. Bosonization techniques [3, 4] then provide a route for the description of the long-range asymptotics of basic correlations. One-dimensional quantum physics is also famous for possessing a class of integrable theories for which all eigenstates can be exactly obtained by Bethe Ansatz [5]. Prototypical examples of Bethe Ansatz solvable models are the Lieb-Liniger model [6,7] of delta-interacting bosons and the Heisenberg spin chain [8,9]. For these and other integrable models, the algebraic Bethe Ansatz [10] and more specifically Slavnov's theorem [11, 12] provides a method to compute matrix elements of local operators between two Bethe states. Summations over relevant matrix elements of particle-hole like excitations can then be implemented, leading to efficient quantitative evaluations of dynamical correlation functions and expectation values of local operators.

The correspondence between results from integrability on the one hand, and Tomonaga-Luttinger theory on the other, are well-known for ground states of the above-mentioned models. The bosonized Tomonaga-Luttinger liquid descriptions are characterized by universal Luttinger parameters related to the compressibility and sound velocity, which can be fitted from the energy levels of the model at finite system size computed by alternative computational methods [13]. For example, the compressibility is readily fitted by the energy levels calculated from integrability upon addition or removal of particles from the Fermi sea. Moreover, analysis of the system size scaling of matrix elements of Umklapp states [14–17] calculated by algebraic Bethe Ansatz methods allows for the determination of non-universal exponents and prefactors appearing in correlation functions of the bosonized theory.

It is interesting to ask whether these correspondences between exact results and field-theory predictions are strictly limited to the vicinity of the ground state, or whether other regions of Hilbert space can similarly be 'captured' both by integrability and by an appropriate field theory. From integrability, things look promising: since the Bethe Ansatz does not care whether the wave functions one writes are near or far from the ground state (these remain exact, irrespective of which energy they have), correlations on such high-energy states can in principle be obtained by extensions of existing ground state-based summation methods. On the other hand, the applicability of Tomonaga-Luttinger theory relies on the linearity of the dispersion relations of the effective excitations around the given state, which cannot be generic for high-energy states. Here, we concentrate on a class of states for which this bosonization procedure can be applied. The setup is easy to visualize: starting from the ground state's Fermi sea configuration, we give a finite opposite momentum to two groups of particles, in fact splitting the Fermi sea in two seas [18], yielding what we coin a 'Moses' state which, while having a macroscopic energy above the ground state, still possesses a zero entropy density (and thus

a potential for displaying critical power-law behavior in its correlations. The corresponding multi-component Tomonaga-Luttinger liquid theory and its correlations were recently obtained for the Lieb-Liniger model [19,20]. The entanglement entropy of these states is nontrivial [21] and consistent with predictions from the effective conformally invariant field theory.

The physical motivation to study such states with a double Fermi sea with opposite momentum kicks originates from experiments where a Bragg pulse is applied to a gas of interacting bosons to create an initial state with counterpropagating particles, leading to a quantum realization of the famous Newton's cradle [22]. It has recently been shown that late-time correlations in the quantum Newton's cradle experiment are better described by a finite entropy version of these states, which can be constructed in the Lieb-Liniger model by theoretically mimicking the application of a Bragg pulse on the ground state and for which the peaks are smoothened and show the characteristic ghost-shaped momentum distribution function at late times [23]. The zero-entropy nature of the states we consider here translates into a quasi-condensate momentum distribution with sharp peaks at finite momenta similar to the ones observed in cold atoms after domain-wall melting of a one-dimensional Mott insulator [24]. At present it is not clear whether a connection exists between the steady state for this quantum quench and the split-Fermi-sea states we consider. Finally, another motivation is related to spin ladders, where states with a Fermi sea consisting of distinct pockets can appear as the ground state [25, 26]. The comparison of their work with ours puts into sharp focus which characteristics should be attributed to the out-of-equilibrium nature of the state and which should not.

The aim of the present article is to extend the approach for the Bose gas with double Fermi seas elaborated in Ref. [20] to the anisotropic Heisenberg spin chain. We compute the dynamical structure factor for this class of excited zero-entropy states from integrability. The static spin-spin correlations are subsequently studied from both the matrix element summation approach from algebraic Bethe Ansatz and the multi-component Tomonaga-Luttinger predictions supported by parameter fitting from integrability, which show surprisingly good correspondence for both the long-range asymptotics and short distances. Going further, we also extend recent advances in the computation of time-dependent correlations by means of mobile impurity models and apply these extensions here. The correspondence with the effective field theory becomes more delicate in this case, but if the separation between seas is large enough the method still gives adequate results.

This article is structured in the following way. Section 2 describes the setup from Bethe Ansatz of the zero-entropy critical states consisting of a state with two Fermi seas, while the dynamical structure factor of these states are evaluated using algebraic Bethe Ansatz matrix element summations in section 3. Section 4 gives the multi-component Tomonaga-Luttinger liquid approach to the real space correlations, which are compared to the algebraic Bethe Ansatz results in section 5. Finally, a description of dynamical correlations by a mobile impurity model is given in section 6.

## 2 Zero-entropy critical states in the XXZ model

The Hamiltonian of the anisotropic Heisenberg spin chain (XXZ model) in a longitudinal magnetic field is given as [8,9]

$$H = J \sum_{j=1}^{N} \left[ S_j^x S_{j+1}^x + S_j^y S_{j+1}^y + \Delta \left( S_j^z S_{j+1}^z - \frac{1}{4} \right) - h S_j^z \right], \tag{1}$$

with periodic boundary conditions $S_{N+1} = S_1$. We fix $J\Delta > 0$ such that the ground state in zero field is antiferromagnetic, and furthermore restrict our analysis to the quantum critical

cases with $\Delta = 1$ and $|\Delta| < 1$. The Tomonaga-Luttinger theory is applicable to these regimes, allowing for a comparison of both approaches.

The commutation of the total spin operator along the $z$-axis $S^z_{\text{tot}} = \sum_{j=1}^N S^z_j$ with Hamiltonian (1) assures a splitting of the Hilbert space in sectors of fixed magnetization $M$. In such a fixed-$M$ subsector, the Bethe Ansatz wave functions [5, 9] are constructed from the fully polarized vacuum state $|0\rangle = \otimes_{j=1}^N |\uparrow_j\rangle$ as plane waves of magnons

$$|\{\lambda\}\rangle = \sum_{j_1 < ... < j_M} \sum_Q A_Q(\{\lambda\}) \prod_{a=1}^M e^{ij_a p(\lambda_{Q_a})} S^-_{j_a} |0\rangle, \tag{2}$$

for which the amplitudes $A_Q(\{\lambda\})$ are set by the scattering phases. Each eigenstate of Hamiltonian (1) is specified by a unique, non-coinciding set of rapidities $\{\lambda\}$ satisfying Bethe equations, which are derived from the scattering phases upon the interchange of two magnons and the imposition of periodic boundary conditions. In logarithmic form, the Bethe equations are given by

$$\theta_1(\lambda_j) - \frac{1}{N} \sum_{k \neq j}^M \theta_2(\lambda_j - \lambda_k) = \frac{2\pi}{N} J_j, \tag{3}$$

where $\theta_n(\lambda) = 2\text{atan}(2\lambda/n)$ for $\Delta = 1$ and $\theta_n(\lambda) = 2\text{atan}(\tanh \lambda / \tan(n\zeta/2))$, $\zeta = \text{acos}(\Delta)$ for $|\Delta| < 1$. The momentum in Eq. (2) is expressible as $p(\lambda) = \pi - \theta_1(\lambda)$.

The Bethe quantum numbers $J_j$ are (half-odd) integers for $N + M$ (even) odd, and form the starting point of the construction of Bethe states at finite system size. The set of rapidities $\{\lambda\}$ can be obtained by solving the logarithmic Bethe equations numerically by an iterative procedure for a given set of non-coinciding quantum numbers $\{J\}$. With the set of rapidities of a Bethe state at hand, properties of the state such as its energy can be calculated directly,

$$E = -\frac{J \varphi_\Delta}{2} \sum_{j=1}^M \theta'_1(\lambda_j) - h\left(\frac{N}{2} - M\right), \tag{4}$$

with $\varphi_\Delta = 1$ for $\Delta = 1$ and $\varphi_\Delta = \sin(\zeta)$ for $|\Delta| < 1$, while the momentum of a Bethe state is expressed directly in terms of its quantum numbers as

$$q = \pi M - \frac{2\pi}{N} \sum_{j=1}^M J_j. \tag{5}$$

In general, rapidities can take on complex values, while the full set $\{\lambda\}$ solving Bethe equations must remain self-conjugate, leading to an arrangement of the rapidities in terms of string solutions. String solutions will not be considered for the initial state consisting of two separate Fermi seas of real rapidities. However, string solutions can occur in the intermediate states of the matrix element summations described in section 3.

The sets of quantum numbers specifying the Bethe states are bounded by limiting quantum numbers $J^\infty$, which are derived by taking the limit of one of the rapidities to infinity and computing the associated quantum number from the Bethe equations (3). The maximum allowed quantum number to still give a finite valued rapidity is then $J^{\max} = J^\infty - 1$. For a Bethe state consisting of $M$ finite real rapidities, one obtains $J^\infty = \frac{1}{2}(N - M + 1)$ and $J^{\max} = \frac{1}{2}(N - M - 1)$, meaning there are $2J^{\max} + 1 = N - M$ available possibilities to distribute $M$ quantum numbers.

The ground state in both zero and finite magnetic field consists of real rapidities [27], where the magnetization sector $M$ (and therefore also the number of rapidities) is fixed by

the magnetic field, with the field strength $h$ acting as a Lagrange multiplier. The quantum numbers of the ground state are

$$J_j^{\text{GS}} = -\frac{M+1}{2} + j \quad \text{for } 1 \leq j \leq M. \tag{6}$$

At zero field ($M = N/2$), the ground state is the only state with real, finite rapidities, implying there is no room for magnon-like excitations in the quantum numbers. In order to apply a shift in the quantum numbers of the Fermi sea, one therefore has to resort to finite field, since there are now many more possibilities for the quantum numbers available beyond the occupation of the Fermi sea alone.

We define the state $|\Phi_s\rangle$ with separated Fermi seas at magnetization $M$ by applying a shift $s$ to the ground state quantum numbers as (we take $M$ to be even)

$$J_j^{\Phi_s} = \begin{cases} J_j^{\text{GS}} - s \text{ if } 1 \leq j \leq \frac{M}{2}, \\[2mm] J_j^{\text{GS}} + s \text{ if } \frac{M}{2} < j \leq M. \end{cases} \tag{7}$$

The effect of shifting the quantum numbers, separating the Fermi sea, is illustrated in Fig. 1. We like to call such a state a Moses state.

$$\{J_j^{\text{GS}}\} \quad \circ \circ \circ \circ \bullet \bullet \bullet \bullet | \bullet \bullet \bullet \bullet \circ \circ \circ \circ$$

$$\{J_j^{\Phi_s}\} \quad \circ \circ \bullet \bullet \bullet \bullet \circ \circ | \circ \circ \bullet \bullet \bullet \bullet \circ \circ$$

Figure 1: Illustration of the distribution of quantum numbers for the ground state (top) and the state with split Fermi sea (bottom) at finite magnetic field. The size of the gap between the two Fermi seas is $2s$ holes.

Four Fermi points are identified, labeled by $i, j, k = 1, 2$ for the left and right sea, and $a, b, c = 1, 2$ for the left or right Fermi point respectively. Furthermore, Fermi momenta are defined by $k_{ia} = \frac{2\pi}{N} J_{ia}$, while signs for each left or right Fermi point in a sea are defined as $s_1 = -1$, $s_2 = 1$. With these notations, generalization to $n > 2$ separated Fermi seas is straightforward.

# 3  Dynamical structure factor

The dynamical structure factor constitutes an important connection between inelastic neutron scattering experiments and theory for quantum spin systems. It is directly related to the differential cross section from scattering experiments on the one hand, while it is computable theoretically from both analytical and (exact) numerical methods. The dynamical structure factor is defined as the Fourier transform of the spin-spin correlation

$$S^{a\bar{a}}(q, \omega) = \frac{1}{N} \sum_{j,j'}^{N} e^{-iq(j-j')} \int_{-\infty}^{\infty} dt \, e^{i\omega t} \langle S_j^a(t) S_{j'}^{\bar{a}}(0) \rangle_c, \tag{8}$$

where the label $a = z, \pm$ distinguishes the longitudinal and transversal structure factors respectively. Transverse structure factors have been computed using field theoretical methods in Ref. [28].

For spin chains at zero temperature, the expectation value in the expression for the spin-spin correlation is evaluated with respect to the ground state of Hamiltonian (1). In the context

of computations for the state with split Fermi sea, the reference state is taken to be the state $|\Phi_s\rangle$ defined from the quantum numbers given by Eq. (7).

The dynamical structure factor can be evaluated by inserting a resolution of the identity in Eq. (8), such that the correlator turns into a sum over matrix elements of the split Fermi sea state and its excitations,

$$S^{a\bar{a}}(q,\omega) = 2\pi \sum_\alpha |\langle \Phi_s | S_q^a | \alpha \rangle|^2 \delta(\omega + \omega_{\Phi_s} - \omega_\alpha). \tag{9}$$

The intermediate states $|\alpha\rangle$ are composed of excitations on the state with double Fermi seas. An adaptive scanning procedure through the most relevant intermediate states is applied using the ABACUS algorithm [29] in order to evaluate the dynamical structure factor. The rapidities of the intermediate states can be obtained by solving Bethe equations (3) with corresponding quantum numbers by an iterative numerical procedure. Subsequently, determinant expressions in terms of the rapidities [30–33] based on Slavnov's formula [11, 12] are employed to evaluate the matrix elements $\langle \Phi_s | S_q^a | \alpha \rangle$. Besides intermediate states consisting of real rapidities, the states $|\alpha\rangle$ can also contain string solutions. In this case, the Bethe-Gaudin-Takahashi equations [34] in terms of string centers are solved, while their matrix elements can be computed by using reduced determinant expressions [33].

By integrating over energy and summing over all momenta for Eq. (8), sum rules for the total intensities of the structure factors can be derived, yielding a quantitative measure on the completeness of the truncated matrix element summations. The sum rules are given by

$$\int_{-\infty}^{\infty} \frac{d\omega}{2\pi} \frac{1}{N} \sum_q S^{zz}(q,\omega) = \frac{1}{4} - \langle S^z \rangle^2, \tag{10}$$

$$\int_{-\infty}^{\infty} \frac{d\omega}{2\pi} \frac{1}{N} \sum_q S^{\pm\mp}(q,\omega) = \frac{1}{2} \pm \langle S^z \rangle, \tag{11}$$

where the magnetization is $\langle S^z \rangle = \frac{1}{2} - \frac{M}{N}$. Sum rule saturations for all ABACUS computation results of dynamical structure factors used throughout this article are listed in Tab. 1. The table displays significantly lower sum rule saturations for the $S^{+-}$ structure factor computations as opposed to the $S^{zz}$ and $S^{-+}$ structure factors. An intuitive explanation for the computational difficulties of $S^{+-}$ is provided in the Bethe Ansatz language, where the operator $S_q^+$ removes a rapidity from the state on which it acts, while the operator $S_q^-$ must add a rapidity to the state. For the latter, it might however be the case that all available quantum number slots around momentum $q$ are already filled, such that much more extensive reorganizations of the state are necessary. Thus, computing $S^{+-}$ requires summing over a much more extensive set of intermediate states than for $S^{-+}$. Given limited computational resources, the saturation of the former will thus be markedly lower than those of the latter.

The longitudinal dynamical structure factors (Eq. (8) with $a = z$) are shown in Fig. 2 for various values of the anisotropy, containing plots for the ground state, as well as for states with a double Fermi sea with varying momentum shifts in the quantum numbers. The transverse dynamical structure factors (Eq. (8) with $a = -$ and $a = +$) have been plotted in Figs. 3 and 4, respectively.

In order to be able to visualize the delta functions in energy appearing in Eq. (9), Gaussian smoothening has been applied to the data as $\delta(\omega) \to e^{-\omega^2/\epsilon^2}/(\sqrt{\pi}\epsilon)$, the width $\epsilon$ being of the order of $1/N$.

The boundaries of the spectra shown in Fig. 2 can be explained by tracking the energies of particle-hole excitations on top of the Moses sea for which either the particle or hole is created at one of the Fermi points. The boundaries of the spectra are equivalent to Fig. 2 in Ref. [20], for the particle-hole excitations for the double Fermi sea in the Lieb-Liniger model.

| | $s$ | $S^{zz}$ | $S^{-+}$ | $S^{+-}$ |
|---|---|---|---|---|
| $\Delta = \frac{1}{10}$ | 0 | 99.50% | 99.50% | 98.74% |
| | 2 | 99.50% | 98.52% | 91.49% |
| | 6 | 99.50% | 97.73% | 89.18% |
| | 12 | 99.16% | 95.27% | 85.47% |
| $\Delta = \frac{1}{2}$ | 0 | 99.50% | 99.50% | 98.97% |
| | 2 | 98.73% | 99.49% | 94.42% |
| | 6 | 98.05% | 99.90% | 90.85% |
| | 12 | 98.00% | 98.16% | 88.61% |
| $\Delta = 1$ | 0 | 99.38% | 99.50% | 94.40% |
| | 2 | 98.04% | 99.50% | 89.45% |
| | 6 | 98.00% | 99.12% | 87.57% |
| | 12 | 97.80% | 98.80% | 86.55% |
| $\Delta = 1$ | $6_l, 18_r$ | 95.67% | 98.01% | |

Table 1: Sum rule saturations for all data obtained from the ABACUS algorithm at $N = 200$ and $M = 50$ for various values of the anisotropy and the momentum shift in the Fermi seas. The bottom row shows the saturations for an asymmetric shift of the quantum numbers.

In particular, for increasing momentum split $s$, the energy of the reference state increases with respect to the ground state, opening up the possibility to populate branches of the spectrum at negative energy. Moreover, the incommensurate points (at zero energy) start moving in momentum. Broad continua of the spectrum remain with sharply defined thresholds, such that the Tomonaga-Luttinger liquid paradigm retains its validity.

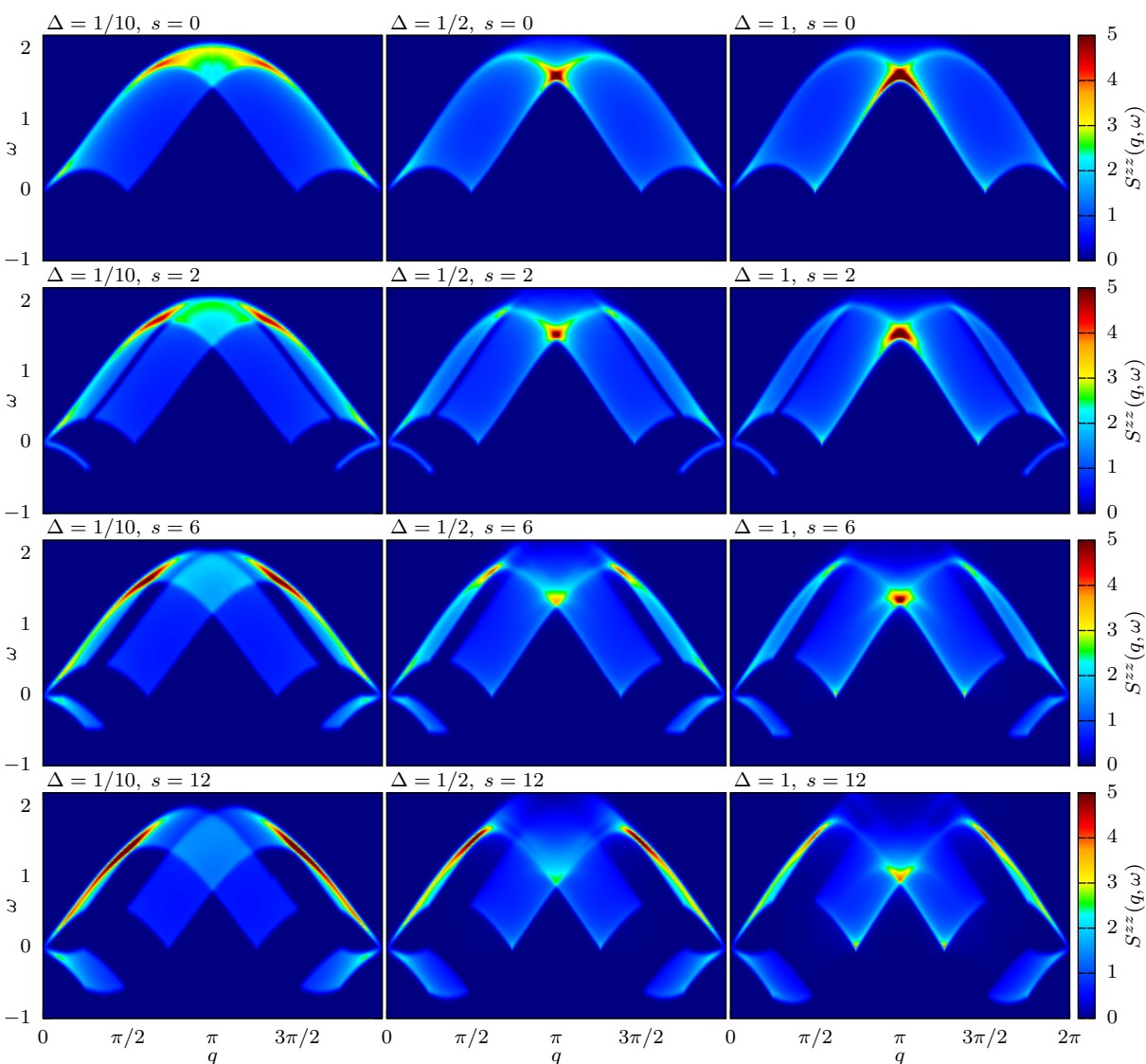

Figure 2: Longitudinal dynamical structure factor $S^{zz}(q,\omega)$ at $N = 200$, $M = 50$ computed from summations of matrix elements obtained from algebraic Bethe Ansatz. From left to right, the anisotropies are $\Delta = \frac{1}{10}, \frac{1}{2}, 1$. From top to bottom, the momentum shifts in the quantum numbers are $s = 0, 2, 6, 12$. The corresponding sum rule saturations of the data are listed in Tab. 1.

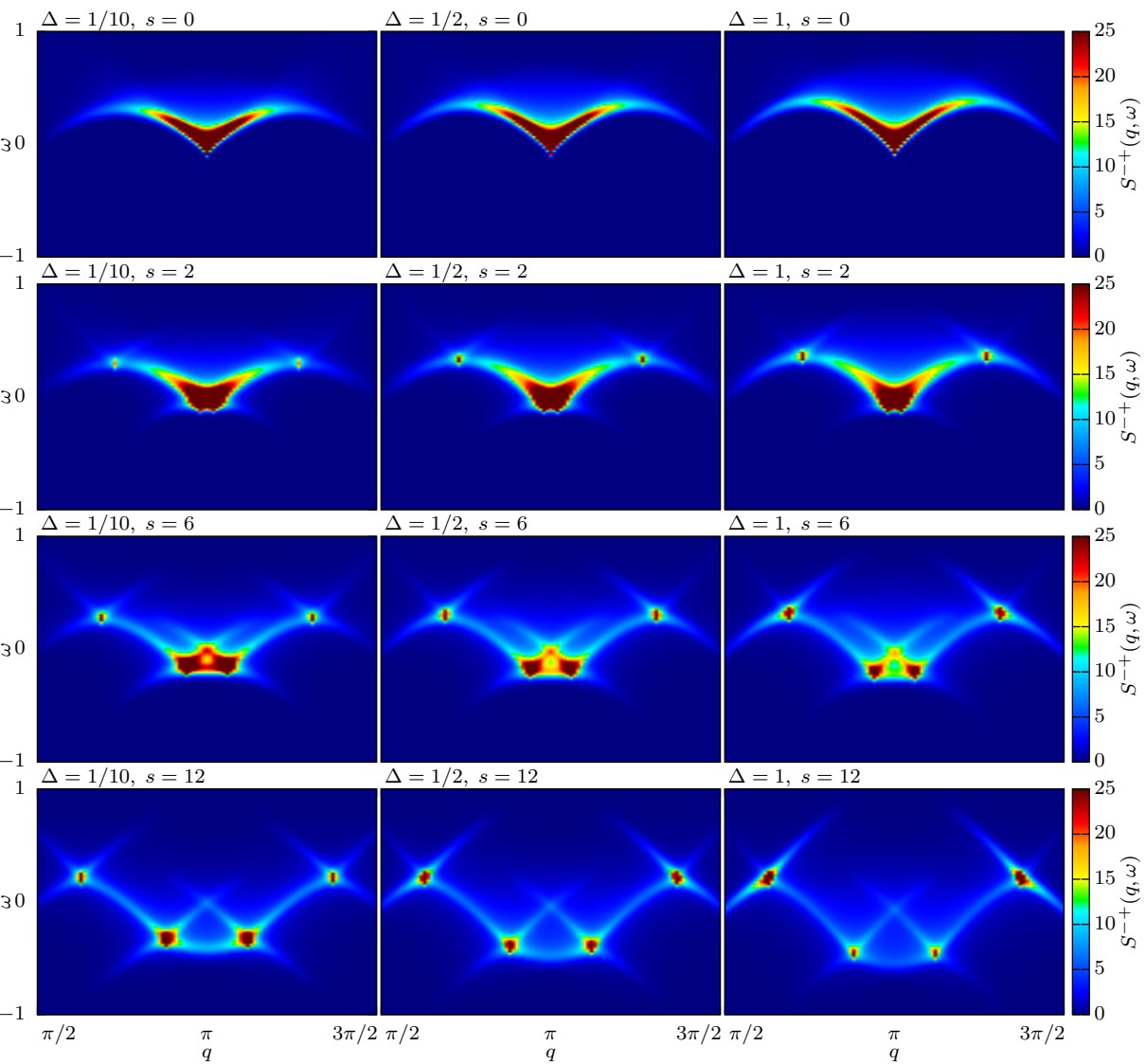

Figure 3: Transverse dynamical structure factor $S^{-+}(q, \omega)$ at $N = 200$, $M = 50$ computed from summations of matrix elements obtained from algebraic Bethe Ansatz. From left to right, the anisotropies are $\Delta = \frac{1}{10}, \frac{1}{2}, 1$. From top to bottom, the momentum shifts in the quantum numbers are $s = 0, 2, 6, 12$. The corresponding sum rule saturations of the data are listed in Tab. 1.

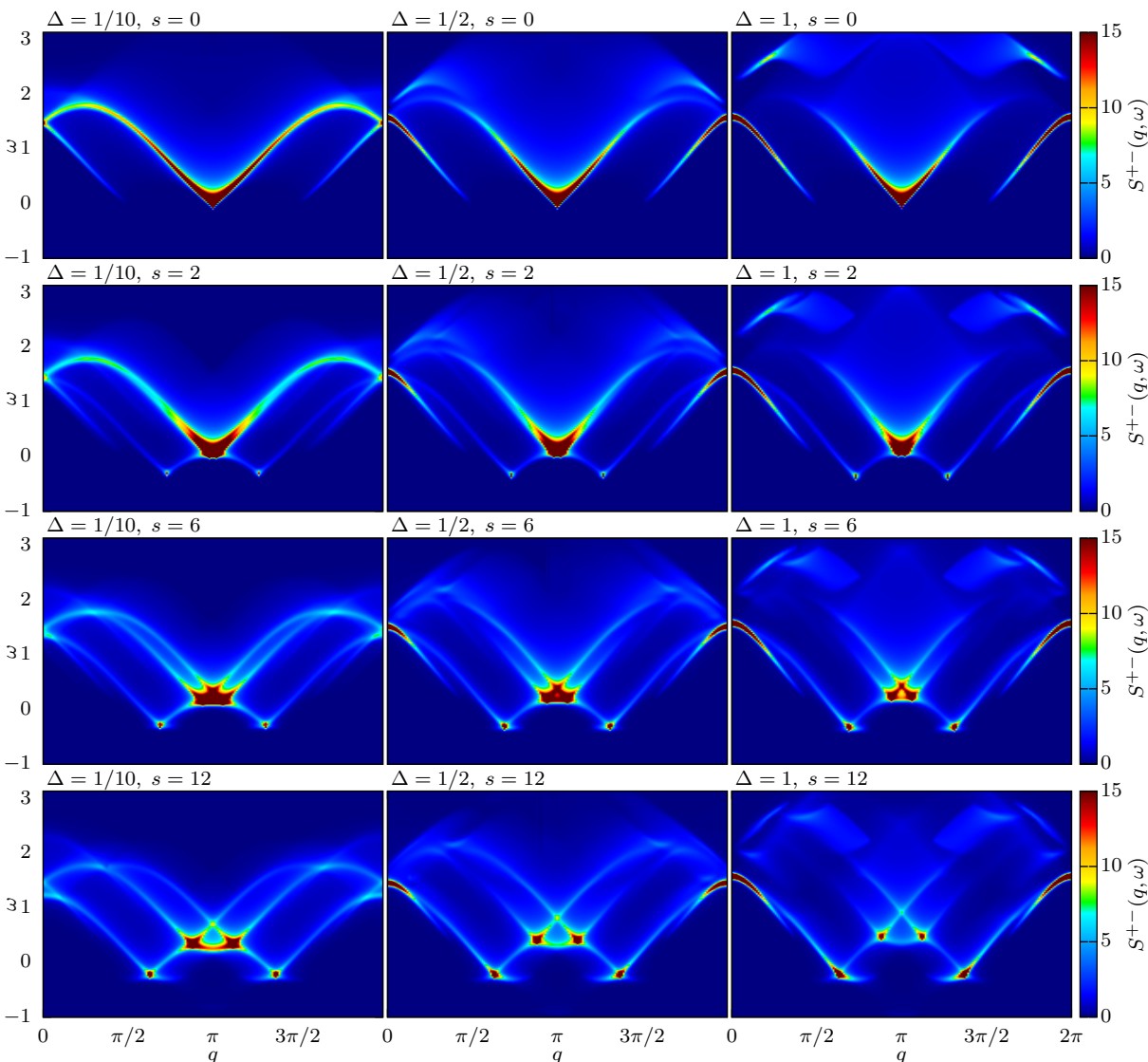

Figure 4: Transverse dynamical structure factor $S^{+-}(q, \omega)$ at $N = 200$, $M = 50$ computed from summations of matrix elements obtained from algebraic Bethe Ansatz, which is furthermore symmetrized around momentum $q = \pi$ in order to obtain higher sum rule saturations with lower computational time. From left to right, the anisotropies are $\Delta = \frac{1}{10}, \frac{1}{2}, 1$. From top to bottom, the momentum shifts in the quantum numbers are $s = 0, 2, 6, 12$. The corresponding sum rule saturations of the data are listed in Tab. 1, which are substantially lower than the other structure factors.

## 4 Multi-component Tomonaga-Luttinger model

The Hamiltonian of the XXZ model can be mapped exactly to spinless fermions on a lattice by the Jordan-Wigner transformation

$$S_j^- \to (-1)^j \cos\left(\pi \sum_{l<j} n_l\right) c_j^\dagger, \quad S_j^+ \to (-1)^j \cos\left(\pi \sum_{l<j} n_l\right) c_j, \quad S_j^z \to \frac{1}{2} - n_j \qquad (12)$$

with $n_j = c_j^\dagger c_j$. The Hamiltonian then reads (neglecting chemical potential terms)

$$H = J \sum_{j=1}^{N} \left[ -\frac{1}{2}\left(c_j^\dagger c_{j+1} + c_{j+1}^\dagger c_j\right) + \Delta n_j n_{j+1} \right]. \qquad (13)$$

The starting point for an effective description of correlations for the split-Fermi-sea states under consideration is the XX point $\Delta = 0$, corresponding to free fermions. The Hamiltonian $H$ is then diagonal in momentum space and our state is characterized by four Fermi points $k_{ia}$.

In constructing a multi-component Tomonaga-Luttinger liquid [35–37] we follow Ref. [20] and introduce a species of chiral fermions $\psi_{ia}$ for each of the Fermi points $k_{ia}$. In the continuum limit, $c_j \to \Psi(x)$, we expand the Jordan-Wigner fermion in terms of the chiral fermions as

$$\Psi(x) \sim \sum_{ia} e^{ik_{ia}x} \psi_{ia}. \qquad (14)$$

This is essentially a mode expansion where each of the chiral fermion species encodes the modes close to $k_{ia}$ which determine the correlations, so that the fields $\psi_{ia}$ can be considered to vary slowly on the scale of the lattice spacing. When $\Delta = 0$, the $\psi_{ia}$ are non-interacting but have nonlinear dispersion. When we switch on interactions but linearize the dispersion relation and neglect backscattering-like terms, the effective Hamiltonian can be written as

$$H_{\mathrm{TL}} = \int dx \left[ \sum_{ia} s_a v_{ia}^0 \psi_{ia}^\dagger (-i\partial_x) \psi_{ia} + \sum_{ia,jb} g_{ia,jb} \rho_{ia} \rho_{jb} \right] \qquad (15)$$

where $\rho_{ia} = \psi_{ia}^\dagger \psi_{ia}$ is the density of species $ia$, $g_{ia,jb}$ are effective $g$-ology-like interaction parameters, $v_{ia}^0 = \partial_k \varepsilon^0(k_{ia})$ are the 'Fermi velocities' corresponding to the bare, cosine dispersion $\varepsilon^0(k) = -J\cos(k)$ of the XX model. Note that we define the velocity by taking the derivative of the dispersion to the right, also at left Fermi points. The combination $s_a v_{ia}^0$ would be positive in equilibrium and corresponding to the Fermi velocity, however, it may be negative in our out-of-equilibrium context. Normal ordering is left implicit.

The validity of Hamiltonian (15) beyond weak interactions to describe correlations cannot be justified by renormalization group arguments in the usual sense since we are describing a high-energy state. Still, the approximations made rely on the idea that we keep the most important terms for the long range physics determined by the modes that are 'close' to the Fermi points $k_{ia}$, which is done by keeping operators of scaling dimension $\leq 2$. We will adhere to the equilibrium terminology and call these marginal while terms of higher scaling dimension are called irrelevant.

Note that by considering the momentum of particle or hole excitations constructed by creating a hole or particle at the Fermi points $k_{ia}$ in quantum number space, it is clear that the $k_{ia}$ do not change when we vary the interaction parameter $\Delta$, as the total momentum is completely determined by the quantum number configuration. We thus observe that a kind of generalized Luttinger's theorem fixes the $k_{ia}$ independent of interactions.

We bosonize the chiral fermions according to

$$\psi_{ia} \sim \frac{1}{\sqrt{2\pi}} e^{-i\sqrt{2\pi}\phi_{ia}}, \qquad \rho_{ia} = -\frac{s_a}{\sqrt{2\pi}} \partial_x \phi_{ia} \tag{16}$$

(where $s_{R,L} = \pm 1$). The Hamiltonian then becomes quadratic in terms of the bosonic fields and can be diagonalized by a Bogoliubov transformation:

$$\phi_{ia} = \sum_{ia,jb} U_{ia,jb} \varphi_{jb}. \tag{17}$$

This results in the diagonal form of the effective Hamiltonian

$$H_{\text{TL}} = \sum_{ia} \frac{s_a v_{ia}}{2\pi} \int dx (\partial_x \varphi_{ia})^2 \tag{18}$$

where the effective velocities $v_{ia}$ are now related to the dressed dispersion of the XXZ model with $\Delta \neq 0$. While the interaction parameters $g_{ia,jb}$ cannot be reliably obtained, the Bogoliubov parameters $U_{ia,jb}$—which also determine the exponents of physical correlation functions—are related to finite-size energy contributions when we extend the filled quantum number blocks by $N_{ia}$ particles at Fermi point $k_{ia}$. The correction $\delta E = E[\{N_{ia}\}] - E[\{0\}]$ is then to order $1/N$ given by

$$\delta E = \sum_{ia} \epsilon_{ia} N_{ia} + \sum_{ia,jb,kc} \frac{\pi}{N} s_c v_{kc} U_{ia,kc} U_{jb,kc} N_{ia} N_{jb}. \tag{19}$$

Here $N_{ia}$ is the number of added (or removed when $N_{ia} < 0$) particles corresponding to chiral species $\psi_{ia}$ and $\epsilon_{ia}$ is the energy associated to Fermi point $k_{ia}$. Eq. (19) gives a relation between the $U_{ia,jb}$ and the finite-size energy differences upon addition or removal of a particle at the Fermi points $k_{ia}, k_{jb}$. Thanks to the properties of the matrix $U_{ia,jb}$, this relation can in fact be inverted and leads to a way to determine $U_{ia,jb}$ and $v_{ia}$ directly from these finite-size corrections. The $U_{ia,jb}$ generalizes the universal compressibility parameter $K$ used in equilibrium situations. In the case of a symmetric combination the derivation may be simplified as detailed in appendix A.

The Bogoliubov parameters $U_{ia,jb}$ have a beautiful interpretation in terms of the phase shifts of the modes at Fermi point $k_{ia}$ upon addition of a particle at Fermi point $k_{jb}$. This can be argued upon refermionization of the effective Tomonaga-Luttinger theory and can be made precise in terms of the shift function $F(\lambda|\lambda')$ describing the change of the rapidities when the system is excited. In the thermodynamic limit the shift function is determined by the integral equation

$$F(\lambda|\lambda') + \sum_j \int_{\lambda_{j1}}^{\lambda_{j2}} d\mu \, a_2(\lambda - \mu) F(\mu|\lambda') = \frac{\theta_2(\lambda - \lambda')}{2\pi} \tag{20}$$

where $a_j(\lambda) = (2\pi)^{-1} \frac{d}{d\lambda} \theta_j(\lambda)$. The relation to the Bogoliubov parameters is then

$$U_{ia,jb} = \delta_{ia,jb} + s_b F(\lambda_{jb}|\lambda_{ia}), \tag{21}$$

which can be shown by comparing the finite size corrections to the energy. A derivation of this relation will be presented elsewhere [38] (also see appendix B). In equilibrium it is known [39] and the shift function plays an important role in going beyond the Luttinger liquid approximation in computing dynamic correlation functions [40, 41].

Physical correlations generally translate into (products of) two-point functions of vertex operators in bosonized language, which in our conventions are evaluated according to

$$\langle e^{i\alpha\sqrt{2\pi}\varphi_{ia}(x)} e^{-i\alpha\sqrt{2\pi}\varphi_{ia}(0)} \rangle = (s_a i/x)^{\alpha^2}. \tag{22}$$

Here, $x$ is measured in units of the lattice spacing.

Asymptotes of spin correlation functions are now obtained by applying the Jordan-Wigner transformation, taking the continuum limit and using the bosonization identities in order to obtain a correlator of the bosonic fields $\phi_{ia}$. The Bogoliubov transformation then expresses this in terms of the free fields $\varphi_{ia}$ for which correlation functions are easily evaluated, leading to an expression involving the $U_{ia,jb}$.

For example, the prediction for the real space longitudinal correlation from the multi-component Tomonaga-Luttinger model is

$$\langle S^z(x)S^z(0)\rangle_{\text{TL}} = s_z^2 - \frac{\sum_{ia,jb,kc} s_a s_b U_{ia,kc} U_{jb,kc}}{4\pi^2 x^2} \tag{23}$$

$$+ \sum_{ia \neq jb} \frac{A_{ia,jb}}{4\pi^2}(-1)^{\delta_{ab}(1-\delta_{ij})} \cos[(k_{ia}-k_{jb})x]\left(\frac{1}{x}\right)^{\mu_{ia,jb}} \tag{24}$$

with

$$\mu_{ia,jb} = \sum_{kc}\left(U_{ia,kc} - U_{jb,kc}\right)^2. \tag{25}$$

Here, non-universal prefactors $A_{ia,jb}$ are included for the fluctuating terms and behave as $1 + \mathcal{O}(\Delta)$. These do not follow from the Tomonaga-Luttinger construction but can be obtained from the finite-size scaling of matrix element detailed in the next section.

The real space transverse correlation is given from the multi-component Luttinger theory as

$$\langle S^-(x)S^+(0)\rangle_{\text{TL}} = \sum_{ia}\sum_{\epsilon=\pm 1} \frac{B_{ia,\epsilon}}{2\pi}(-1)^{\delta_{s_a,\epsilon}} e^{-ik_{ia,\epsilon}x}\left(\frac{1}{x}\right)^{\mu_{ia,\epsilon}} \tag{26}$$

where $k_{ia,\epsilon} = k_{ia} + \epsilon\pi M/N$ and again non-universal prefactors, now denoted $B_{ia,\epsilon}$, are included. The exponents are

$$\mu_{ia,\epsilon} = \sum_{kc}\left[\sum_{jb}(\epsilon/2 + s_a\delta_{ia,kc})U_{jb,kc}\right]^2. \tag{27}$$

In the next section we discuss how all parameters are obtained from numerical evaluation of the spectrum and matrix elements and how these predictions compare to real-space correlations obtained from the ABACUS data.

## 5 Real space correlations

The real space spin-spin correlations of the double Fermi sea state are directly obtained from the ABACUS dynamical structure factor data from section 3 by applying an inverse Fourier transform,

$$\langle S_j^a(t)S_0^{\bar{a}}(0)\rangle = \frac{1}{N}\sum_\alpha |\langle\Phi_s|S_{q_\alpha-q_{\Phi_s}}^a|\alpha\rangle|^2 e^{i(q_\alpha-q_{\Phi_s})j - i(\omega_\alpha-\omega_{\Phi_s})t}. \tag{28}$$

All computations have been carried out at system size $N = 200$ at half-magnetization $S^z = \frac{N}{2} - M$, $M = 50$, for various values of anisotropy and momentum shifts in the Fermi seas. The sum rule saturations of all ABACUS data used in this section are listed in Tab. 1. The results for the static real space correlations ($t = 0$) are plotted as data points in Figs. 7-8 (longitudinal, $a = z$) and Figs. 9-10 (transverse, $a = -$) respectively. The multi-component Tomonaga-Luttinger model predictions for the correlations with fitted parameters from integrability are incorporated in the figures as well. The predictions for real-space correlations

from the transverse structure factor with $a = +$ differ only at $x = 0$ from $a = -$. Although the sum-rule saturation for this correlation (see Tab. 1) is considerably less, we have checked that the fit of the real-space correlation is comparable to the case $a = -$.

The longitudinal real space correlation from Eq. (24) requires the determination of three classes of parameters. First, the parameters $U_{ia,jb}$ can be deduced from the behavior of the energy upon removal or addition of particles to all four Fermi points. We therefore consider the second derivative of the finite size corrections to the Tomonaga-Luttinger model (Eq. (19)) and define a matrix $G_{ia,jb}$ as

$$G_{ia,jb} = \frac{N}{\pi} \frac{\partial^2 E_0}{\partial N_{ia} \partial N_{jb}} = \sum_{kc} s_c v_{kc} U_{ia,kc} U_{jb,kc}. \tag{29}$$

By considering all possible combinations of adding and removing particles to the four Fermi points, the second derivative with respect to energy can be calculated from the energy levels obtained from Bethe Ansatz (Eq. (4)). Subsequently, the eigenvectors of the matrix $G_{ia,jb}$ yield all the parameters $U_{ia,jb}$.

The remaining non-universal prefactors $A_{ia,jb}$ and exponents $\mu_{ia,jb}$ can be obtained using the system size scaling behavior of the Umklapp matrix elements, from the relation

$$|\langle \Phi_s | S_j^z | ia, jb \rangle|^2 = \frac{A_{ia,jb}}{4\pi^2} \left( \frac{2\pi}{N} \right)^{\mu_{ia,jb}}, \tag{30}$$

where $|ia, jb\rangle$ is defined as the Umklapp state by removing a particle at the Fermi point labeled by $ia$ and placing it back at the Fermi point labeled by $jb$. By scaling the system size $N$, the parameters $A_{ia,jb}$ and $\mu_{ia,jb}$ are directly obtained by a linear fit to the logarithm of Eq. (30) for all 6 combinations of Umklapp states. This procedure is repeated for all values of anisotropy and momentum shifts in the quantum numbers considered here. The values of the prefactors and exponents are plotted in Fig. 5 as function of anisotropy for a fixed value of the momentum shift to the Fermi seas.

In order to compare the multi-component Tomonaga-Luttinger model predictions to finite size results, a conformal transformation

$$x \to \frac{N}{\pi} \sin(\pi x / N) \tag{31}$$

is applied to the scaling behavior of Eq. (24). The resulting expressions for the longitudinal real space correlations, along with the parameters obtained by the procedure described above, are plotted in Figs. 7 and 8. Fig. 7 shows the correspondence of the multi-component Tomonaga-Luttinger model to the matrix element summations obtained from ABACUS at $N = 200$, $M = 50$, for a momentum shift in the Fermi seas of $s = 12$. For all distances but the very smallest, both approaches show good agreement. Fig. 8 displays a comparison at very short distances for different momentum shifts and anisotropy, still showing a large agreement in both approaches, in a regime where the Tomonaga-Luttinger model is not a priori expected to give bonafide predictions.

Similar to the determination of the non-universal prefactors and exponents for the longitudinal case, the scaling relation

$$|\langle \Phi_s | S_j^- | ia, \epsilon \rangle|^2 = \frac{B_{ia,\epsilon}}{2\pi} \left( \frac{2\pi}{N} \right)^{\mu_{ia,\epsilon}} \tag{32}$$

allows for the determination of the parameters $B_{ia,\epsilon}$ and $\mu_{ia,\epsilon}$ for the purpose of Eq. (26). The Umklapp state $|ia, \epsilon\rangle$ is defined by the removal of a particle at the Fermi point labeled by $ia$, while $\epsilon = \pm$ dictates the direction of the shift in the quantum numbers due to change

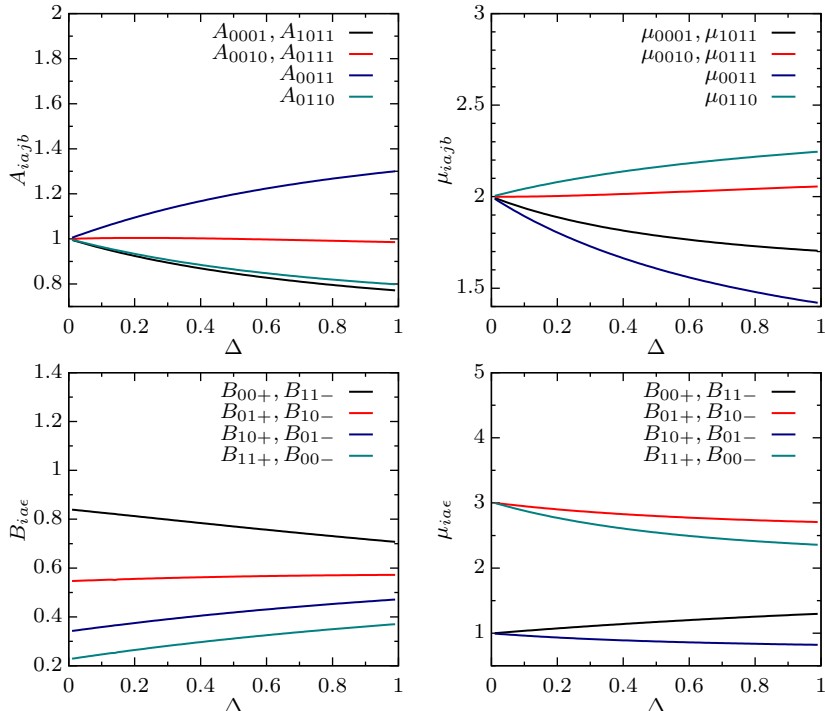

Figure 5: Non-universal prefactors (left column) and exponents (right column) as a function of anisotropy for the static real space $\langle S^z(x)S^z(0)\rangle$ (top row) and $\langle S^-(x)S^+(0)\rangle$ (bottom row) correlations of the multi-component Tomonaga-Luttinger model, computed from integrability for a state with a double Fermi sea at $N = 200$, $M = 50$ and momentum shift $s = 12$.

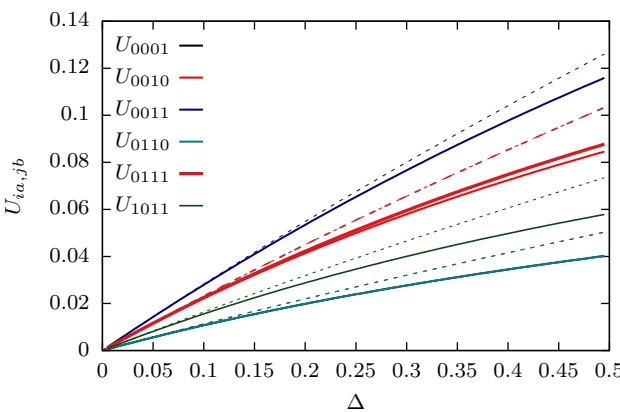

Figure 6: Bogoliubov parameters as a function of anisotropy for the static real space $\langle S^z(x)S^z(0)\rangle$ correlations of the multi-component Tomonaga-Luttinger model. The figure shows a comparison between the expansion in small anisotropy (dashed lines) and the parameters computed from integrability (solid lines) for a state with a double Fermi sea at $N = 200$, $M = 50$ and momentum shift $s = 12$.

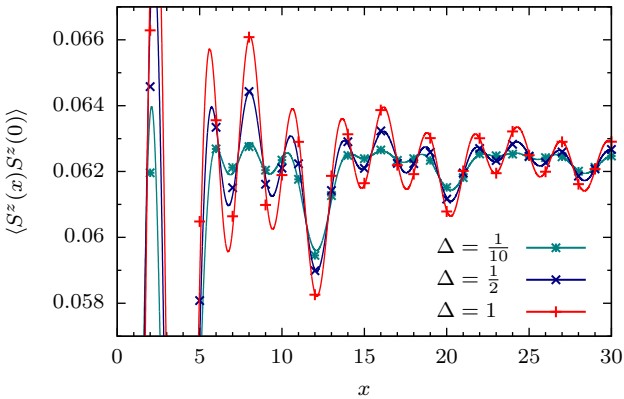

Figure 7: Longitudinal static real space correlation $\langle S^z(x)S^z(0)\rangle$, at $N = 200$, $M = 50$ for several values of the anisotropy, for a state with momentum split in the Fermi sea by $s = 12$. The figure compares the ABACUS results (points) to the multi-component Tomonaga-Luttinger model (lines). The agreement holds not only at large distances, but also down to very short ones.

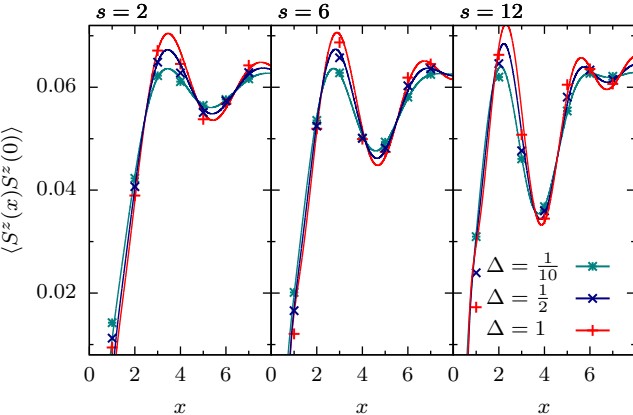

Figure 8: Longitudinal static real space correlation $\langle S^z(x)S^z(0)\rangle$ for very short distances, at $N = 200$, $M = 50$ for several values of the anisotropy, for states with a momentum split in the Fermi sea by $s = 2, 6, 12$ (from left to right panels respectively). The figure compares the ABACUS results (points) to the multi-component Tomonaga-Luttinger model (lines).

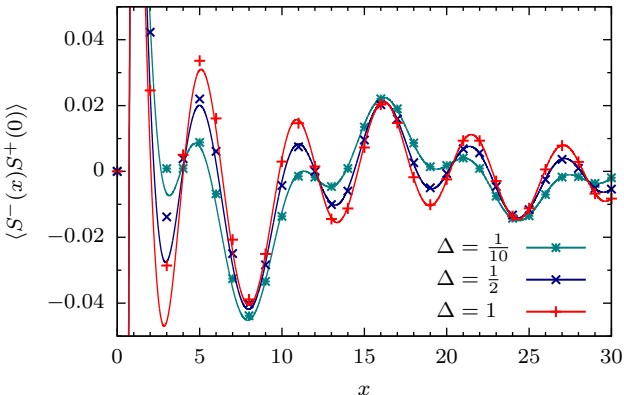

Figure 9: Transverse static real space correlation $\langle S^-(x)S^+(0)\rangle$, at $N = 200$, $M = 50$ for several values of the anisotropy, for a state with momentum split in the Fermi sea by $s = 12$. The figure compares the ABACUS results (points) to the multi-component Tomonaga-Luttinger model (lines). The correspondence works well for distances larger than a handful of sites.

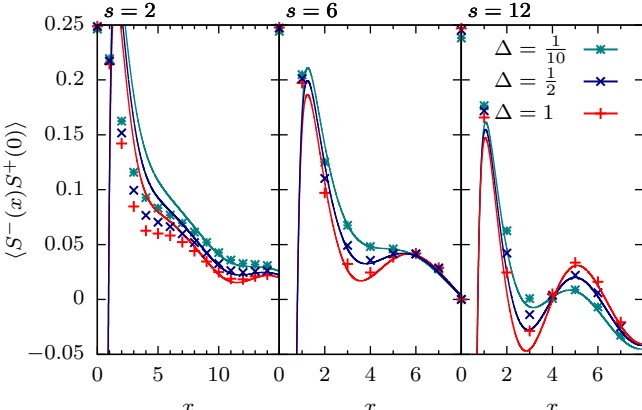

Figure 10: Transverse static real space correlation $\langle S^-(x)S^+(0)\rangle$ for very short distances, at $N = 200$, $M = 50$ for several values of the anisotropy, for states with a momentum split in the Fermi sea by $s = 2, 6, 12$ (from left to right panels respectively). The figure compares the ABACUS results (points) to the multi-component Tomonaga-Luttinger model (lines).

in the parity after changing the number of particles. Again, the prefactors and exponents are obtained by fitting the finite size scaling behavior of Eq. (32) and their values are plotted in Fig. 5 as function of anisotropy for a fixed value of the momentum shift to the Fermi seas.

The transverse real space correlations from Eq. (26) are plotted for several values of the anisotropy in Fig. 9 for a fixed value of the momentum shift and in Fig. 10 for short distances and three separate values of the momentum shift, respectively. The non-universal prefactors and exponents are obtained by the aforementioned method, while the conformal transformation to finite size from Eq. (31) has been applied as well. Both figures show again perfect agreement for large distances, while the agreement is also good for short distances with respect to system size. The smallest momentum shift ($s = 2$ in Fig. 10) shows the worst agreement at very short distances.

Finally, all previous procedures have been applied to a state where an asymmetric momentum shift is employed to separate the Fermi seas. Fig. 11 shows the corresponding longitudinal and transverse dynamical structure factors and real space correlations obtained from ABACUS at $N = 200$, $M = 50$, and the multi-component Tomonaga-Luttinger model. The parameters for the latter have been obtained from the fitting procedure described in this section, applied to system size scaling behavior of Umklapp matrix elements on the asymmetric Fermi points. Once again, the real space correlations display agreement for both the asymptotics as well as for short distances for both approaches.

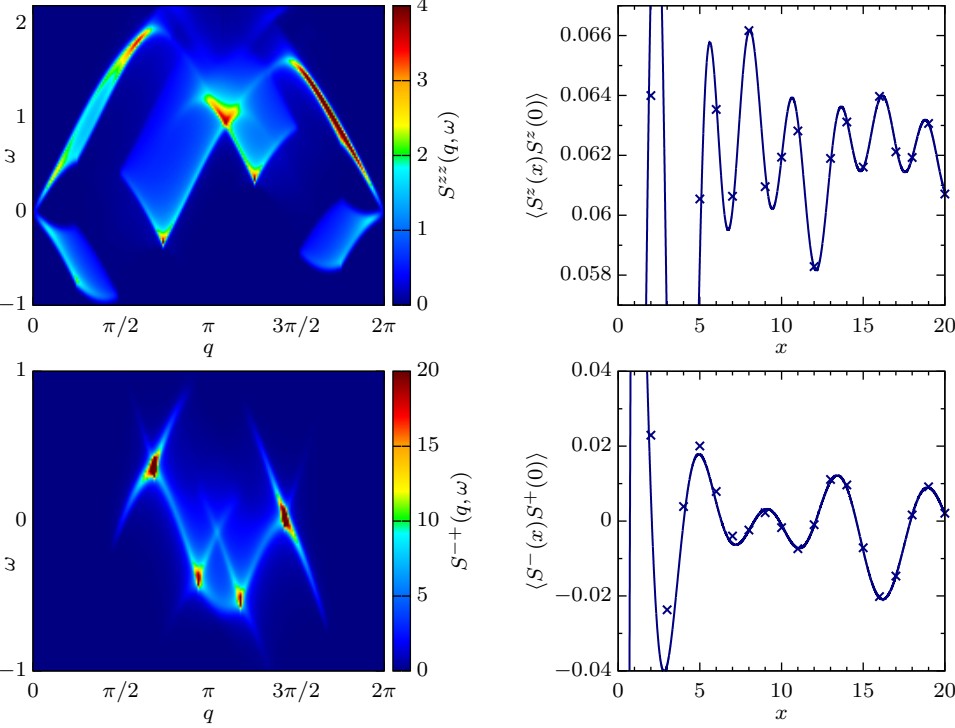

Figure 11: Dynamical (left column) and static (right column) correlations for a state with an asymmetric momentum shift for the two Fermi seas by $s_l = 6$, $s_r = 18$, at $N = 200$, $M = 50$, $\Delta = 1$. The panels in the right column show the resemblance between the real space correlations obtained from ABACUS (points) and the multi-component Tomonaga-Luttinger model (lines). The saturation of the ABACUS data is given in Tab. 1.

# 6 Time dependence of correlations

The fact that the zero-entropy states we are considering are far from equilibrium is not visible when we restrict attention to the static correlations, which would be the same if this state was obtained as the ground state of a different Hamiltonian. In order to make the out-of-equilibrium nature apparent we have to probe the energies of 'excitations', i.e. modifications of the Moses sea by creating additional particles and holes, which may now have both positive and negative energy differences with respect to the reference state. A physically meaningful way to probe the energy landscape is by computing time-dependent correlations which can in principle be related to observable quantities. These are already encoded in the dynamical structure factors computed with ABACUS and can be obtained by Fourier transformation.

Recent years have witnessed a revolutionary increase in understanding of dynamical correlations in critical one-dimensional systems from the perspective of both effective field theory methods and integrability [17, 40, 42–67]. The threshold behavior of many dynamical correlations in energy and momentum space can be understood in terms of specific configurations of particle and hole excitations. These lead to a scattering phase shift of the modes close to the Fermi energy which is identified as the cause of the characteristic power-law singularities by means of Anderson's orthogonality catastrophe. This threshold behavior, which also determines the asymptotic behavior of the correlations in real space and time, is then described by an effective model in which the high energy particle or hole excitation is treated as a mobile impurity interacting with the low-energy modes.

We generalize this mobile impurity approach to the present out-of-equilibrium context by extending our multi-component Tomonaga-Luttinger model to include the appropriate impurity configurations and interactions. To be specific, we will compute the spin autocorrelation

$$C(t) = \langle S_j^z(t) S_j^z(0) \rangle = \langle \Psi^\dagger(t)\Psi(t)\Psi^\dagger(0)\Psi(0) \rangle, \tag{33}$$

where $\Psi(t) = \Psi(x = 0, t)$ denotes the Jordan-Wigner fermion and we used translational invariance. By imagining to obtain $C(t)$ as a Fourier transform in $(k, \omega)$-space taking the $k$-integral first, one can argue that as a function of $\omega$ singular behavior stems from the 'Fermi points' and points where the edge of support has a tangent with vanishing velocity. This identifies the important impurity configurations for this function as corresponding to a particle or hole with vanishing velocity, i.e. either at the bottom or the top of the band. Let us assume that the Moses state leaves the corresponding quantum numbers unoccupied, which is valid for a symmetric configuration with an even number of seas. This means that there are only high-energy particle impurities. We will use an index $\gamma = 0, 1$ to label the particle at the bottom/top of the band respectively. The mobile impurity model becomes

$$H_{\text{MIM}} = \int dx \left[ \sum_{ia} \frac{s_a v_{ia}}{2} (\partial_x \varphi_{ia})^2 + \sum_\gamma d_\gamma^\dagger \left( \epsilon_\gamma - \frac{\partial_x^2}{2m_\gamma} \right) d_\gamma - \sum_{ia,\gamma} \frac{s_a \kappa_{ia,\gamma}}{\sqrt{2\pi}} d_\gamma^\dagger d_\gamma \partial_x \varphi_{ia} \right]. \tag{34}$$

Note that the last term is just a density-density interaction between the impurity modes and the chiral fermions parametrized by the coupling constants $\kappa_{ia,\gamma}$, while the first two terms correspond to the impurity dispersions and the TL modes. In the small $\Delta$ limit all parameters in $H_{\text{MIM}}$ can be obtained from $H$ in Eq. (13) as in appendix C. In general they can be obtained from integrability.

The impurity modes are decoupled from the Tomonaga-Luttinger model up to irrelevant operators by the unitary transformation [68]

$$U = \exp\left\{ i \int dx \sum_{ia,\gamma} \frac{\kappa_{ia,\gamma}}{s_a v_{ia} \sqrt{2\pi}} d_\gamma^\dagger d_\gamma \varphi_{ia} \right\}. \tag{35}$$

The effect is that correlators of the TL model are still computed in the same way, but the impurity operator obtains an extra vertex operator in terms of the bosonic modes

$$d \rightarrow d \exp\left\{-i \sum_{ia} \frac{\kappa_{\gamma,ia}}{s_a v_{ia} \sqrt{2\pi}} \varphi_{ia}\right\}. \tag{36}$$

The logic is identical to the ground state case, and this also suggests that we can identify the parameter $\kappa_{ia,\gamma}/v_{ia}$ as the phase shift at the Fermi points upon creating the impurity according to [40, 41]

$$\frac{\kappa_{ia,\gamma}}{v_{ia}} = 2\pi F(\lambda_{ia}|\lambda_\gamma). \tag{37}$$

For the computation of the autocorrelation $C(t)$, the crucial observation is now that the asymptotic behavior, determined by the behavior around a few singular points of the longitudinal structure factor, is well captured by certain correlations computable using the Hamiltonian $H_{\mathrm{MIM}}$. In marked contrast to the equilibrium case, the TL model does not account for zero-energy states only, and therefore even the contributions to Eq. (33) that do not involve the impurity will display the energy difference of the Fermi points leading to fluctuating terms

$$e^{i(\epsilon_{ia}-\epsilon_{jb})t} \langle \psi_{ia}^\dagger(t)\psi_{jb}(t)\psi_{jb}^\dagger(0)\psi_{ia}(0)\rangle \tag{38}$$

where $\epsilon_{ia}$ is the energy associated to Fermi point $k_{ia}$. The TL contributions sum up to an expression similar to the static correlation in Eq. (23). The impurity contributions are of the form

$$e^{i(\epsilon_{ia}-\epsilon_\gamma)t} \langle \psi_{ia}^\dagger(t)d_\gamma(t)d_\gamma^\dagger(0)\psi_{ia}(0)\rangle. \tag{39}$$

Using the impurity correlator

$$\langle d_\gamma(t)d_\gamma^\dagger(0)\rangle = \int \frac{dk}{2\pi} e^{-i\frac{k^2}{2m_\gamma}} = \sqrt{\frac{m_\gamma}{2\pi i t}} \tag{40}$$

we find the result

$$C(t) \sim s_0^2 - \sum_{ia,jb,kc} \frac{s_a s_b U_{ia,kc} U_{jb,kc}}{4\pi v_{kc}^2 t^2} + \sum_{ia,jb} \frac{A_{ia,jb} \cos(\epsilon_{ia}-\epsilon_{jb})}{4\pi^2} \prod_{kc}\left(\frac{1}{i s_c v_{kc} t}\right)^{[U_{ia,kc}-U_{jb,kc}]^2}$$
$$+ \sum_{ia,\gamma} \frac{A'_{ia,\gamma} e^{i(\epsilon_{ia}-\epsilon_\gamma)t}}{2\pi} \sqrt{\frac{m_\gamma}{2\pi i t}} \prod_{kc}\left(\frac{1}{i s_c v_{kc} t}\right)^{[U_{ia,kc}+\frac{\kappa_{ia,\gamma}}{2\pi s_c v_{kc}}]^2}. \tag{41}$$

The prefactor $A'_{ia,\gamma} = 1 + \mathcal{O}(\Delta)$ can in principle be obtained from finite-size scaling of matrix elements similar to $A_{ia,jb}$ but with the Umklapp state replaced by the appropriate impurity state [69]. We have checked this expression for the autocorrelation against Fourier transformed ABACUS data for small values of $\Delta$, and find that it converges to the exact result on quite short time scales for a configuration when the Fermi points $k_{ia}$ are well away from the band top and bottom at $k_\gamma = 0, \pi$ (Fig. 12), but the correspondence for short to moderate times becomes noticeably worse when we decrease the separation between the two seas. This could be a finite size effect since the number of states in between the impurity mode and the Fermi edges becomes small, but rather we believe that the correlation is not well-captured by the impurity model in that case as a clear separation in sub-bands becomes questionable.

When the current mobile-impurity approach works well, this tells us that the time-dependent correlation is determined by the modes very close to the Fermi points $k_{ia}$ which correspond to particle-hole excitations involving only the quantum numbers close to the edges of the two Fermi seas. The role of the spectrum at the Fermi points and of the impurity is two-fold:

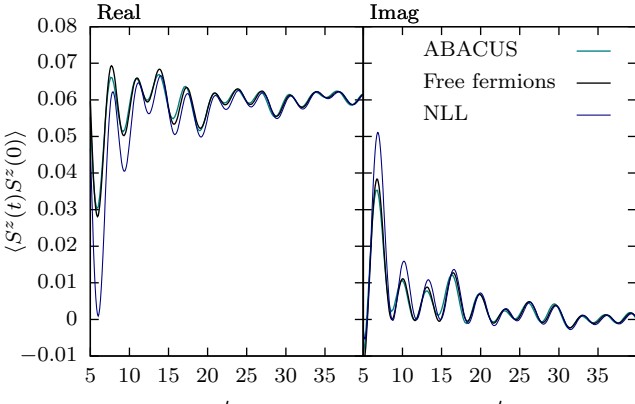

Figure 12: Comparison of computations for the autocorrelations $\langle S^z(t)S^z(0)\rangle$, from ABACUS at $N = 200$, $M = 50$, $\Delta = \frac{1}{100}$, $s = 12$, from free fermions ($\Delta = 0$), and from non-linear Luttinger (NLL) theory with effective field parameters taken for free fermions.

(i) The energy differences determine the frequencies of fluctuations. (ii) The Fermi velocities $v_{ia}$ and impurity mass $m_\gamma$ change the prefactor of the separate terms. Note that the decay of the correlation on the other hand is determined by energy independent data, namely by the appropriate phase shifts and Anderson's orthogonality catastrophe, very similar to the equilibrium case.

# 7 Conclusions

We have considered high energy zero-entropy states for the anisotropic Heisenberg spin chain defined by a double-Fermi sea quantum-number configuration. Our focus was on dynamical correlations computed by summing over relevant matrix elements of particle-hole excitations at finite system size, the matrix elements being given by algebraic Bethe Ansatz. Correlations in real space and time are obtained from these by numerical Fourier transformation.

Zero-entropy Bethe states provide an interesting class of eigenstates of integrable models which, while far from equilibrium, share many features with the ground state. In particular we have shown that, when in the critical regime of the anisotropic Heisenberg spin chain, these states display critical correlations characterized by fluctuations with power-law decay. Starting from the Bethe Ansatz solution, we have constructed an effective field theory in terms of a multi-component Tomonaga-Luttinger model capturing these correlations with great accuracy for large, but also surprisingly short, distances in real space. Similar to the ground state case, fluctuations correspond to the differences of the generalized Fermi momenta $k_{ia}$, the power-law exponents are related to a Bogoliubov transformation diagonalizing the effective Hamiltonian, while non-universal prefactors determine the relative amplitudes of the fluctuating terms.

The Bogoliubov parameters $U_{ia,jb}$ can be related to finite-size energy differences upon adding or removing particles at the Fermi points, and the non-universal correlation prefactors to the finite-size scaling of matrix elements with Umklapp excitations. We have used this to obtain all field theory parameters from integrability yielding parameter-free fits for the static correlations. A surprising fact is that, while static correlations do not know about the Hamiltonian for time evolution, the energies computed to obtain the $U_{ia,jb}$ do feature the Fermi velocities from the spectrum of the actual XXZ Hamiltonian, which does not determine the statistical ensemble in the present case. This implies that the relation of the spectrum and the Bogoliubov parameters is universally valid irrespective of the energy function one uses.

The parameters $U_{ia,jb}$ turn out to correspond to the phase shifts of the modes at one of the generalized Fermi points $k_{jb}$ upon creating an excitation at Fermi point $k_{ia}$. At zero temperature, the Luttinger parameter $K$ can similarly be regarded as a parametrization of the phase

shift at the left and right Fermi points upon creating single-particle excitations with momentum $\pm k_F$, and it has recently been realized that this implies a universal description of dynamical correlations that go beyond the linear-spectrum approximation made in the Tomonaga-Luttinger model [48]. The applicability of the multi-component Tomonaga-Luttinger model suggests that, similar to the ground state case, a universal description of dynamical correlations in a domain close to the Fermi points may be obtained. The description of domains near the edge of support is likely captured by a mobile impurity model. As a first step of this generalized use of what is known as non-linear Luttinger liquid theory, we have compared the longitudinal autocorrelation function to predictions obtained from a mobile impurity model at small $\Delta$. The correlation seems to converge to the mobile impurity result in quite reasonable times at least when the two Fermi seas are well-separated.

In conclusion, we have presented results on dynamical correlations of zero-entropy states in the anisotropic Heisenberg chain. The distinctive features, which may serve for identification in experiment, can be understood by adapting the familiar ground state reasoning. By making the appropriate adjustments to equilibrium techniques based on the Tomonaga-Luttinger model many aspects can furthermore be understood in great detail and with quantitative agreement once a handful of parameters is fixed from integrability. Although we have focused on states which have vanishing entropy density, we may consider thermal-like dressings to the split seas. In the Tomonaga-Luttinger description, finite temperatures are treated by a simple functional replacement for the fundamental correlators. On the side of integrability, recent work has shown that finite temperature correlators are also numerically accessible, at least for the Lieb-Liniger model [70]. How the correspondence between integrability results and the field theory works out in split-sea configurations at finite temperature remains to be investigated.

# Acknowledgements

We thank Yuri van Nieuwkerk for useful comments. The authors acknowledge support from the Foundation for Fundamental Research on Matter (FOM) and from the Netherlands Organization for Scientific Research (NWO). This work forms part of the activities of the Delta-Institute for Theoretical Physics (D-ITP).

**Funding information** The work of R. P. V. and J.-S. C. was supported by NWO VICI grant number 680-47-605. The work of I. S. E. and J.-S. C. was supported by FOM program 128, 'The singular physics of 1D electrons'.

# A    Simplification for symmetric seas

In the case of a symmetric configuration of $n$ seas, we have $-k_{iL} = k_{n+1-iR}$ and $-v_{iL} = v_{n+1-iR}$ and in general that the system is symmetric under simultaneously exchanging $i \leftrightarrow n+1-i$ and $L \leftrightarrow R$. It is then convenient to combine the fields at Fermi points at opposite momenta and define

$$\phi_i = \frac{1}{\sqrt{2}}(\phi_{iL} - \phi_{n+1-iR}), \qquad \theta_i = \frac{1}{\sqrt{2}}(\phi_{iL} + \phi_{n+1-iR}). \tag{42}$$

with inverse transformation

$$\phi_{iL} = \frac{1}{\sqrt{2}}(\theta_i + \phi_i), \qquad \phi_{n+1-iR} = \frac{1}{\sqrt{2}}(\theta_i - \phi_i). \tag{43}$$

The Hamiltonian for the multi-component TL model can then be written as

$$H_{\mathrm{TL}} = \sum_i \frac{v_i^0}{2} \int dx [(\partial_x \phi_i)^2 + (\partial_x \theta_i)^2] + \sum_{ij} \int dx \frac{1}{2\pi} [g_{ij}^+ \partial_x \phi_i \partial_x \phi_j + g_{ij}^- \partial_x \theta_i \partial_x \theta_j] \tag{44}$$

with $v_i^0 = v_{n+1-iR}^0$ and $g_{ij}^\pm = g_{iLjL} \pm g_{n+1-iRjL}$.

In order to respect the canonical commutation relations and diagonalize $H_{\mathrm{TL}}$ we introduce new fields according to

$$\phi_i = \sum_j U_{ij} \varphi_j, \qquad \theta_i = \sum_j [U^{-1}]_{ji} \vartheta_j \tag{45}$$

where the $U_{ij}$ are related to the Bogoliubov parameters $U_{ia,jb}$ as

$$U_{ij} = U_{iLjL} - U_{iLn+1-jR} \tag{46}$$

by virtue of the symmetry $U_{ia,jb} = U_{n+1-i\bar{a},n+1-j\bar{b}}$. The effective Hamiltonian takes the familiar diagonal form

$$H_{\mathrm{TL}} = \sum_i \frac{v_i}{2} \int dx \left[ (\partial_x \varphi_i)^2 + (\partial_x \vartheta_i)^2 \right]. \tag{47}$$

The matrix $U_{ij}$ is straightforwardly obtained from calculations in finite size and finite particle number from corrections to the energy upon creating particle-number or current excitations

$$\delta E = \sum_i \frac{\pi v_i}{2L} \left[ \left( \sum_j [U^{-1}]_{ij} \Delta N_j \right)^2 + \left( \sum_j U_{ji} \Delta J_j \right)^2 \right]. \tag{48}$$

Here

$$\Delta N_i = N_{n+1-iR} + N_{iL}, \qquad \Delta J_i = N_{n+1-iR} - N_{iL} \tag{49}$$

in terms of the numbers $N_{ia}$ of particles added at Fermi point $k_{ia}$.

## B  Moses states and generalized TBA

Finite-size corrections to the spectrum for the ground state are usually discussed by taking the thermodynamic limit of the Bethe equations and the energy, keeping finite size corrections using the Euler-Maclaurin formula [10]. The class of Moses states can be considered as the zero temperature limit of a generalized Gibbs ensemble for which a generalized thermodynamic Bethe Ansatz and a treatment of finite-size corrections similar to the ground state exists [71, 72]. However, in our case we still use the energy dispersion obtained from the true XXZ Hamiltonian determining the time-evolution rather than the statistical ensemble (which we define by hand in the micro-canonical sense). Since we also obtain the critical exponents using the true XXZ Hamiltonian and not a GGE Hamiltonian, the fact that the results coincide requires clarification. The standard derivation of the finite size corrections of the energy makes use of the property that the corresponding dressed energy vanishes at the Fermi points, but in our case $\epsilon(\lambda_{ia}) \neq 0$ if we measure energies by using the true XXZ Hamiltonian.

Upon changing the particle number $N_{ia}$ at one of the Fermi points $k_{ia}$ the extremal rapidities $\lambda_{ia}$ experience a shift of order $1/L$. The change in the Fermi points and rapidities can be shown to satisfy

$$\delta k_{ia} = -s_a N_{ia} \frac{2\pi}{L}, \tag{50}$$

$$\delta \lambda_{ia} = \sum_{jb} \frac{1}{Ls_a \rho(\lambda_{ia})} [\delta_{ia,jb} + s_a F(\lambda_{ia}|\lambda_{jb})] N_{jb}. \tag{51}$$

The finite-size corrections to the energy due to adding or removing particles from the extremities of the Fermi-sea-like blocks is then

$$\delta E = \sum_{ia}\epsilon(\lambda_{ia})N_{ia} + \frac{1}{N}\sum_{ia,jb,kc}\frac{s_c\epsilon'(\lambda_{kc})}{2\rho(\lambda_{kc})}[\delta_{ia,kc}+s_cF(\lambda_{kc}|\lambda_{ia})][\delta_{jb,kc}+s_cF(\lambda_{kc}|\lambda_{jb})]N_{ia}N_{jb}$$

(52)

where $\pm\epsilon(\lambda)$ is the energy of a particle (hole) with rapidity $\lambda$ on top of the Moses state. In contrast to the equilibrium case where energy is measured or a GGE dressed energy satisfying $\epsilon_{\text{GGE}}(\lambda_{ia}) = 0$, the funtion $\epsilon(\lambda)$ cannot be defined through an integral equation but has an additional contribution stemming from the finite energy at the Fermi points. See [38] for details.

## C  Perturbative expressions for effective-field-theory parameters

A convenient way to obtain perturbative expressions for the exponents of asymptotes of correlation functions is to derive the effective Hamiltonian to first order in the interaction. Starting from the Hamiltonian of spinless fermions on the lattice we use the mode expansion

$$\Psi(x) \sim \sum_{ia}e^{ik_{ia}x}\psi_{ia}(x) + \sum_{\gamma}e^{ik_{\gamma}x}d_{\gamma}(x)$$

(53)

and bosonize the chiral fermions. The kinetic term leads to the velocities for the chiral fermions $v_{ia}^0 = J\sin(k_{ia})$ and the impurity parameters in the noninteracting limit $\epsilon_{\gamma}^0 = \mp J$ and $m_{\gamma}^0 = \pm J$. The interaction term

$$H_{\text{int}} = \sum_{x}J\Delta n(x)n(x+1)$$

(54)

renormalizes these values. By plugging in the bosonization identities, normal ordering and neglecting irrelevant terms we get the first order in $\Delta$ expressions

$$g_{ia,jb} = J\Delta \begin{cases} \sum_{ld}\cos(k_{ia}-k_{ld}), & (ia = jb) \\ 1-\cos(k_{ia}-k_{jb}), & (ia \neq jb). \end{cases}$$

(55)

These give

$$U_{ia,jb} = \delta_{ia,jb} - [1-\delta_{ia,jb}]\frac{J\Delta}{\pi}\frac{s_b[1-\cos(k_{ia}-k_{jb})]}{v_{ia}^0 - v_{jb}^0}.$$

(56)

Next, we focus on the terms from the interaction involving the impurity. This leads to a density-density interaction of the impurity modes with the chiral fermions with parameters

$$\kappa_{ia,\gamma} = 2J\Delta[1-\cos(k_{ia}-k_{\gamma})].$$

(57)

There is also a first order correction to the impurity energy appearing from Eq. (54) from the terms proportional to $d^{\dagger}d$ (after normal ordering), which is

$$\epsilon_{\gamma=0,1} = \mp J\left(1\mp 2n_0\Delta + \sum_{ia}\frac{\Delta}{\pi}s_a\sin(k_{ia})\right).$$

(58)

This corresponds to the Hartree-Fock correction

$$\delta\epsilon_{\gamma} = \sum_{i}\int_{k_{iL}}^{k_{iR}}\frac{dk}{2\pi}[V(0)-V(k_{\gamma}-k)]$$

(59)

with $V(q) = 2J\Delta\cos(q)$, which corresponds to the interaction potential in Eq. (54): $H_{\text{int}} = \frac{1}{2L}\sum_{q}V(q)n_q n_{-q}$.

The non-universal prefactors can also be obtained perturbatively using the methods discussed in Ref. [16], but we have not done this calculation.

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
