# Peer review of "Correlations of zero-entropy critical states in the XXZ model: integrability and Luttinger theory far from the ground state"

_SciPost Physics, doi:SciPost Phys. 1, 008 (2016)_

## Round 1 · Referee Report · Anonymous · 2016-7-20

Strengths

1-Very well written
2-Clear exposition of the results
3-Nice physical discussions throughout the manuscript

Weaknesses

I don't see weaknesses

Report

In this manuscript the authors consider spin-spin correlations of a special class of highly excited states in the gapless regime of the XXZ Heisenberg spin-chain. In terms of rapidity distribution functions, these states are described by a splitted Fermi sea and while being highly excited, they display properties similar to those of the ground-state, such as a vanishing Yang-Yang entropy.

The authors computes dynamical structure factors using the integrability based ABACUS algorithm, from which real space and time correlators are obtained exactly. These results are compared with the predictions of a multi-component Tomonaga-Luttinger model, where all the free parameters are fixed from energetics and finite-size scaling of matrix elements computed via Bethe ansatz. While TL predictions are expected to be accurate for asymptotics of real space spin-spin correlators, the authors findings also explicitly show that the agreement between the latter and Bethe ansatz results extend to very short values of the distance between the spins.

The paper extends to the XXZ Heisenberg chain the analysis of analogous states in the Lieb-Liniger model, as reported in Ref. [20]. Here additional technical difficulties arise in the computation of correlators, due to the presence of states displaying strings, which have to be considered in the spectral decompositions of the identities during numerical evaluations with the ABACUS algorithm.

The manuscript is very clear and well written. Furthermore the results are nicely presented, with interesting physical discussions.

Requested changes

I don't request any changes to the manuscript, and only have two minor questions/remarks.

1-First, from Table 1, as explicitly noted by the authors, one can see that the saturation rules for S^+- are significantly worse than the case S^-+. While one could imagine that different excitations are important for transverse and longitudianl correlators, one could naively expect that similar excitations have to be considered for the two aforementioned transverse correlators. Table 1 show that this is not the case: do the authors have an intuition for this disparity between transverse correlators, or is this only justifiable a posteriori?

2-Second, a trivial comment: in the main text I didn’t find any reference to the appendices, which appear a bit disconnected w. r. t. the main text. The authors might consider to add a reference to the appendices in the parts of the manuscript where the corresponding results are used. Analogously, in the last part of the introduction, section 6 and 7 (together with the appendices) are not mentioned in the plan of the paper.

  • validity: top
  • significance: high
  • originality: good
  • clarity: top
  • formatting: perfect
  • grammar: perfect

Author:  Rogier Vlijm  on 2016-09-06  [id 53]

(in reply to Report 1 on 2016-07-20)

We thank the referee for the report and the remarks on our manuscript. We have formulated a response and will resubmit our manuscript.

Requested changes:
1 - The referee asks a very pertinent question concerning the saturation of the $S^{+-}$ correlator as compared to that of the $S^{-+}$ one. Indeed the computation of $S^{+-}$ is numerically much more demanding than that of $S^{-+}$. The reason can be intuitively explained as follows: in the Bethe language, the operator $S^+$ removes a rapidity from the state on which it acts (we assume positive magnetization as in our submission, and a reference state with all spins up). If we consider the $q$ momentum Fourier component of this spin raising operator, the rapidity that will be removed is situated in the vicinity of this momentum (of course accompanied by further dressings and multiparticle corrections coming from interactions). The action of $S^+_q$ is thus either more or less to project to zero (if there is no rapidity around momentum $q$) or equal to simply removing the rapidity at momentum $q$ (if it is present on the state).

In contrast, the operator $S^-_q$ adds a rapidity around momentum $q$ to the state. It might however be the case that all available quantum number slots around momentum $q$ are already filled; in this case, it becomes like trying to add a fermion in the middle of an already-filled Fermi sea: the only way to do this is to create much more extensive reorganizations of the state. Thus, computing $S^{+-}$ requires summing over a much more extensive set of intermediate states than for $S^{-+}$. Given limited computational resources, the saturation of the former will thus be markedly lower than those of the latter.

We shall add this reasoning to an updated version of our manuscript.  

2 - We thank the referee for pointing out these missing sections in the plan of the paper and the absence of mentioning the appendices, and incorporated the references into our manuscript.

---

## Round 1 · Referee Report · Anonymous · 2016-7-28

Strengths

1- All the results are discussed in a clear and interesting way.
2- Impressive agreement between field theory predictions and exact Bethe ansatz results.

Weaknesses

No weaknesses.

Report

This manuscript deals with static and dynamical correlations in excited states of anisotropic spin chains that correspond to split Fermi seas. In previous work (Ref. [20]), two of the authors studied the equivalent problem for the Bose gas. They showed that split-Fermi-sea states can be investigated by adapting Bethe ansatz and field theory methods developed to compute ground state correlations. In this work the analysis is extended to an integrable lattice model; a noteworthy complication in this case is that string solutions have to be taken into account in the form factor summation.

The authors have performed a careful and detailed analysis of their results. In particular, the agreement between the analytical expressions for equal-time correlations and the result from the numerical computation of form factors corroborates the validity of their methods. The manuscript should definitely be published. I only have a few comments and requested changes that I would like the authors to consider.

Requested changes

1- I believe the lower index q_\alpha in Eq. (9) should be replaced by the momentum q in S(q,\omega). The same q_\alpha appears in Eq. (28), but there the authors should explain that it refers to the difference between the momenta of the states \Phi_s and \alpha.

2- Please comment that the velocities v_{ia} defined in Eq. (15) carry a sign depending on the Fermi point (otherwise one might be confused by assuming that these parameters are always positive).

3- Unlike Figs. 2 and 4, the momentum axis in Fig. 3 does not cover the entire Brillouin zone, but only runs between pi/2 and 3pi/2. In this case the axes labels are a bit confusing because the 3pi/2 in one plot overlaps with pi/2 in the next plot.

4- The results for S^{+-} with Delta=1 and s>0 in Fig. 4 look strange because the spectral weight at higher energies (omega>2) is not symmetric about q=pi, even though the Fermi seas are split symmetrically. Since this is the case of worst saturation of the sum rule in Table 1, one might worry that some contributions could be missing here.

5- The transverse dynamical structure factors S^{+-} and S^{-+} were studied by H. Karimi and I. Affleck in Phys. Rev. B 84, 174420 (2011). The authors should cite this reference.

6- Please clarify whether the coefficients A’_{ia,\gamma} in Eq. (41) can be determined exactly from finite size scaling, like the other coefficients in Eqs. (30) and (32), or if they are treated as free parameters when comparing with the ABACUS result in Fig. 12.

7- In section 6 the authors apply the mobile-impurity approach to study time-dependent correlations, but limit their discussion to small values of Delta. Figure 12 only shows the result for the free fermion case Delta=0. Previous applications of this approach to equilibrium correlations were also checked for strongly interacting models. Is there any reason for restricting to Delta=0 here?

  • validity: top
  • significance: high
  • originality: good
  • clarity: top
  • formatting: excellent
  • grammar: perfect

Author:  Rogier Vlijm  on 2016-09-06  [id 54]

(in reply to Report 2 on 2016-07-28)

We thank the referee for the report and detailed inquiries concerning our manuscript.

Requested changes:
1 - The lower index should indeed be replaced by the momentum, and the Eq. (28) should be dependent on the momentum difference between the states \Phi_s and \alpha, as \Phi_s can have arbitrary momentum. We thank the referee for spotting these details in our definitions.
2 - A comment about the sign of v_{ia} added below Eq. (15).
3 - A label for 3\pi/2 has been added to the figures, such that the labels do not overlap anymore.
4 - The asymmetry of the results displayed in the first version of our manuscript can indeed be fully attributed to the missing contributions, supported by the relatively low sum rule saturations of the S^{+-} correlations. The symmetric shift of the quantum numbers should give back a fully symmetric structure factor. In order to achieve a higher sum rule saturation, the results can be demanded to be symmetric, and run the computation only on intermediate states with N/2 <= P < N, therefore reducing the computational efforts needed. We have updated the correlations for $\Delta=1$ in Fig. 4 using this approach. Morerover, the S^{+-} correlations are generally much harder to assess computationally, see also our comment on Report 6.
5 - A citation to this reference has been added.
6 - The parameters A’_{ia,\gamma} can in principle be computed from the ABACUS finite system size scaling of matrix elements. However, specifically in Fig. 12, they are taken =1 appropriate for the free fermion point.
7 - There is no principle reason for restricting to \Delta =0. While the formulas for the mobile impurity model are valid for arbitrary \Delta, we have decided to present only this case as a first simple check in this preliminary study as the quantities in the effective field theory are straightforward to obtain.

---

## Editorial Decision

published